# Determinants of self-reported health status during COVID-19 lockdown among surveyed Ecuadorian population: A cross sectional study

Iván Dueñas-Espín[1]*, Constanza Jacques-Aviñó[2,3], Verónica Egas-Reyes[4], Sara Larrea[5], Ana Lucía Torres-Castillo[1], Patricio Trujillo[1], Andrés Peralta[1]

1 Instituto de Salud Pública, Facultad de Medicina, Pontificia Universidad Católica del Ecuador, Quito, Ecuador, 2 Fundació Institut Universitari per a la Recerca a l'Atenció Primària de Salut Jordi Gol i Gurina (IDIAPJGol), Barcelona, Spain, 3 Universitat Autònoma de Barcelona, Bellaterra, Cerdanyola del Vallès, Barcelona, Spain, 4 Facultad de Psicología Pontificia Universidad Católica del Ecuador, Quito, Ecuador, 5 Independent Consultant, Quito, Ecuador

* aaperalta@puce.edu.ec

## Abstract

### Objective

To examine the associations of sociodemographic, socioeconomic, and behavioral factors with depression, anxiety, and self-reported health status during the COVID-19 lockdown in Ecuador. We also assessed the differences in these associations between women and men.

### Design, setting, and participants

We conducted a cross-sectional survey between July to October 2020 to adults who were living in Ecuador between March to October 2020. All data were collected through an online survey. We ran descriptive and bivariate analyses and fitted sex-stratified multivariate logistic regression models to assess the association between explanatory variables and self-reported health status.

### Results

1801 women and 1123 men completed the survey. Their median (IQR) age was 34 (27–44) years, most participants had a university education (84%) and a full-time public or private job (63%); 16% of participants had poor health self-perception. Poor self-perceived health was associated with being female, having solely public healthcare system access, perceiving housing conditions as inadequate, living with cohabitants requiring care, perceiving difficulties in coping with work or managing household chores, COVID-19 infection, chronic disease, and depression symptoms were significantly and independently associated with poor self-reported health status. For women, self-employment, having solely public healthcare system access, perceiving housing conditions as inadequate, having cohabitants requiring care, having very high difficulties to cope with household chores, having COVID-19, and having a chronic disease increased the likelihood of having poor self-reported health

**Data Availability Statement:** All relevant data are within the paper and its Supporting Information files.

**Funding:** The authors received no specific funding for this work.

**Competing interests:** The authors have declared that no competing interests exist.

status. For men, poor or inadequate housing, presence of any chronic disease, and depression increased the likelihood of having poor self-reported health status.

## Conclusion

Being female, having solely public healthcare system access, perceiving housing conditions as inadequate, living with cohabitants requiring care, perceiving difficulties in coping with work or managing household chores, COVID-19 infection, chronic disease, and depression symptoms were significantly and independently associated with poor self-reported health status in Ecuadorian population.

## Introduction

Globally, lockdown's impact on physical and mental health during the COVID-19 pandemic has been well-documented [1–6]. In several countries worldwide, lockdowns cause a high incidence of anxiety, depression, post-traumatic stress disorder, psychological distress, and other types of stress in the general population [7–9]. These conditions seem to affect women, children, adolescents, the elderly, and populations experiencing socioeconomic deprivation [7–10]. However, few studies have examined the specific impact of lockdown measures in the context of rampant social and economic inequalities and weak states with low emergency response capacities.

Specifically, in the Latin American and Caribbean regions, accentuated impacts from lockdown and the COVID-19 pandemic have generated several undesired effects such as the formation of COVID-19 hotspots exacerbated by weak social protection structures, fragmented health systems, and deep inequalities [11]. Although the region's development process was facing serious structural limitations before the pandemic, it is expected that COVID-19 will cause a worse recession in the region, with subsequent contraction of regional gross domestic product (GDP) [12].

Similar to other Latin American countries, Ecuador was severely affected by the COVID-19 pandemic [13]. In 2020, the country reached the second highest rate of confirmed cases in South America [10] and the ninth place worldwide in the number of deaths per million people [14]. In April 2020, Ecuador's case called global attention to disturbing images of corpses piling up in the streets of Guayaquil [15]. In the same year, 24% of the urban population and 49% of the rural population lived under the poverty line [16]. Moreover, only approximately 30% of the economically active population had an adequate job, that is, employed persons who, during a reference week, received labor income equal to or greater than the minimum wage and worked equal to or greater than 40 hours a week, regardless of the desire and availability to work additional hours [17]. Fewer than 60% of rural households had access to the Internet, and fewer than 20% owned a computer [18].

Despite the precarious economic situation of most of the population, the measures taken by the Ecuadorian government to prevent COVID-19's spread were mainly restrictive interventions, such as social distancing policies and mandatory lockdowns enforced by the police and military forces [13]. The Ecuadorian government's response to the global emergency was also characterized by poor epidemiological surveillance, lack of access to the public health system, corruption scandals, null community participation, and insufficient social support [19, 20]. In June 2021, the Ecuadorian government implemented a plan for massive vaccinations.

Thus, for more than a year, people living in Ecuador had to cope with the fear of contracting COVID-19, compounded by movement and rights restrictions, an education system that

was only allowed to operate online, a collapsed health system unable to provide care for both COVID-19 and other health conditions, and an economic recession [21]. Moreover, despite some social protection measures, such as food kits adopted by local governments, NGOs, churches, and civil society organizations, many basic needs went unfulfilled owing to lockdown restrictions, such as access to medicines for chronic patients [22]. Several fields of public health and social care were neglected by Ecuadorian authorities, leading to a profound worsening of health across all social classes, and the management of social risks, exacerbated by the pandemic, fell to families, particularly women. In this sense, the impact has been especially significant for women, children, teenagers, and people with disabilities [23].

These limited economic and social aid policies have resulted in significant reductions in the coverage of health supplies and medications for chronic patients, maternal health, and sexual and reproductive health as well as unequal vaccine access [24]. In this context, a few studies conducted in Ecuador reported an increased prevalence of psychological distress symptoms [25]. Some studies and anecdotal data point to an increase in many factors that are determinants of poor mental health during a health emergency, such as the burden of unpaid domestic labor for women, gender-based violence [13, 19], child abuse [26], lack of access to medicine for prior chronic diseases, poor housing conditions, and overcrowding [27].

After a brief revision from literature in Medline, we found several papers studying the social impact from the COVID-19 lockdown in Ecuadorian population. Regarding its impact on lifestyles, one study [28] found that teachers were not ready for the sudden shift to emergency remote teaching. Another study [29] found that stress was associated with poorer diet quality. Therefore, the confinement affected various areas of the lives of citizens.

Regarding the impact of lockdown on mental health of Ecuadorian population, one study [30] found that burnout has a mediating effect between job motivation and turnover intention, and that female and male workers' burnout and turnover intentions levels are different when intrinsic motivation is present. Otherwise, a multicenter study [31] showed that the higher perception of stress, the less self-care activities are adopted, and in turn the lower the beneficial effects on well-being.

Regarding knowledge, attitudes and practices towards COVID-19, a paper [32] found that participants reported high levels of adoption of preventive practices; importantly, unemployed individuals, househusbands/housewives, or manual laborers, as well as those with an elementary school education, have lower levels of knowledge about COVID-19.

In the mental health area, a paper [33] found that cognitive emotion regulation strategies on anxiety and depression was moderated by the sex of participants and the time of assessment. Moreover, a study [34] found that age was significantly correlated with all the psychological variables; importantly, females presented higher levels of stress, especially those who have home care responsibilities.

Thus, despite that several papers have been published, it is not clear how the determinants of self-reported and mental health affected the general population during lockdown; and, specifically, how the lockdown circumstances affected to men and women differentially.

Self-reported health status is an indicator of people's health, the use of health services, and mortality [35]. Moreover, scientific evidence supports the idea that self-reported health differs according to social class and job insecurity [36], with important differences by sex and gender [37]. It is therefore relevant to analyze confinement's effects on self-perceived health, particularly because of its potential to explain and interpret inequities as explanatory factors of health in the lockdown context, as has been shown by other studies in the region [1].

Therefore, and to examine the association between self-reported health status and its associated factors during Ecuador's COVID-19 lockdown, we conducted a cross-sectional survey of between July and to October 2020 to adults who were living in Ecuador between March to

October 2020. As a secondary objective, we aimed to assess the differences in these associations between women and men.

## Materials and methods

### Design, population, and sample

This was a cross-sectional study based on an online survey (**see S1 File**). Participants were recruited through online platforms and social media using convenience and snowball sampling. Participants were aged 18 years or older and lived in Ecuador between March and October 2020.

### Survey and measurements

We conducted a cross-sectional survey of between July and to October 2020 to adults who were living in Ecuador between March to October 2020. The survey was created by a group of experts including psychologists, statisticians, and epidemiologists, and was previously applied in different countries as part of a wider study carried out by a group of researchers from the *Institut Universitari d'Atenció Primària* IDIAPJGol (Spain), FIOCRUZ Brasília, Brazil, from the School of Public Health of the University of Chile, and from the Instituto National Public Health Mexico, School of Public Health of Mexico. Data were collected using the Survey Monkey® platform hosted by IDIAPJGol. The survey questions were worded according to the cultural and language particularities of Ecuador.

### Dependent variable

The dependent variable was self-reported health status, with five response options on a Likert scale, which was then dichotomized into good self-perceived health (very good and good) and poor self-perceived health (fair, poor, and very poor), which has been used for similar purposes in other studies [1, 27, 35].

### Independent variables

We employed the following independent variables in the survey: participants' demographics (sex, age, and location); socioeconomic status (education level, employment status, access to health services), living conditions (housing area, total number of cohabitants, number of cohabitants who require care, age of cohabitants, and perception of the type of housing's adequacy for lockdown, suffering violence (during the lockdown), difficulties in coping with work or managing household chores, health-oriented behaviors (physical activity during lockdown as well as alcohol, cigarette, illicit drugs, and sugary drinks consumption); COVID-19 related experiences and perceptions (having had COVID-19, degree of concern of being infected with SARS-CoV-2), having chronic diseases, and general health status (prior chronic illnesses, use of medicines).

Anxiety was measured using the Generalized Anxiety Disorder Scale (GAD-7) [38] and was categorized as normal, mild, moderate, and severe, and depression was assessed using the Patient Health Questionnaire (PHQ-9) [39] and was categorized as none/minimal, mild, moderate, and moderately severe.

### Statistical analyses and sample considerations

Assuming an alpha risk of 5% and a beta risk of 20%, it was necessary to recruit at least 1235 individuals to estimate with a confidence level of 95% and a precision of +/- 1.5 percentage units; a population percentage having fair or poor general health will predictably be around

7% [40]. The necessary replacement percentage was predicted to be 10%. We employed the GRANMO sample calculator version 7.12 [41].

Descriptive statistics were performed using percentages for categorical variables and medians and interquartile ranges (IQR) for discrete and non-normally distributed variables. We tested normality by checking the histograms. We performed $Chi^2$ to compare differences in proportions of explanatory variables across the two categories of health self-perception and the U-Mann-Whitney test to assess differences in discrete or non-normally distributed explanatory variables across health self-perception categories. We then estimated the crude and adjusted odds ratios (aOR) of regular or poor self-perception of health status for each explanatory variable and its categories.

We then fitted multivariate logistic regression models to evaluate the independent association between each explanatory variable (age, sex, education level, educational level, employment status, access to health services, social security, perception of the type of housing's adequacy for lockdown, housing area, number of cohabitants who require care, physical activity during lockdown, alcohol consumption, degree of concern of being infected with SARS--CoV-2, difficulties in coping with the job or taking care of household chores, healthy or socially-active activities during lockdown, violence or abuse during lockdown, diseases, symptoms and medications, anxiety, depression, and use of antidepressants), and health status self-perception. First, we built a saturated model that included all individual covariates. Then, based on the researchers' criteria, we eliminated covariates with Wald test p-value>0.25 from the model [42], and 95% confidence intervals (95%CI) of the aOR and their corresponding p-values were calculated. Once the parsimonious model was obtained, we compared both models and chose the "final" model, according to its level of significance from the likelihood ratio test. Considering that the percentage of missing data was <23%, we employed a complete case analysis to estimate statistical associations (for further details see **S1 Table**).

Analyses were stratified by sex; respondents who had a non-binary gender identity or did not identify with other categories were excluded from the analysis, because the group was too small (n = 4). Sex was tested as an effect modifier and a confounder.

To test for potential effect modification, we performed several secondary analyses to assess the sensitivity of our estimates with our assumptions regarding biases as well as to test for model misspecifications. We ran the final model excluding *(i)* high- and low-educated subjects, *(ii)* those with chronic diseases, *(iii)* those with severe anxiety, and *(iv)* those with severe depression.

Statistically significant differences were considered when the p-value was <0.05; all analyses were performed using Stata 16.1 (*Statistical Software Stata*: *Release 16.1 College Station*, *TX*: *StataCorp LP*).

## Ethics approval

This study was approved by the Research Ethics Committee on Human Beings (CEISH) of the Ministry of Public Health of Ecuador (code number MSP-CGDES-2020-0129-O) with the authorization to obtain an online informed consent before the start of the survey. Minors were not included in the study.

## Results

### Descriptive results

We analyzed the information of 2924 people. The participant characteristics are presented in **Table 1**. Their median (IQR) age was 34 (27–44) years, and most patients were female (68%).

**Table 1. Description of the sample.**

| Variable[a] | Whole sample n = 2924 |
|---|---|
| Age in years of life, median (IQR) | 34 (27 to 44) |
| Female, n % | 1801 (68) |
| Education level | |
| *Educational level lower than university education, n %* | 408 (16) |
| *University educational level or higher, n %* | 2191 (84) |
| Employment status | |
| *Public or private full job, n (%)* | 1601 (63) |
| *Self-employment, n (%)* | 315 (13) |
| *Unpaid work, retired or student, n (%)* | 612 (24) |
| Access to health services | |
| *Social security[b], n (%)* | 1517 (61) |
| *Private health insurance, n (%)* | 598 (24) |
| *Public health services user, n (%)* | 375 (15) |
| Perception of the adequacy of the type of housing to lockdown | |
| *Moderately to well adequate, n (%)* | 2204 (86) |
| *Little or not adequate, n (%)* | 348 (14) |
| Housing area | |
| *<50 $m^2$, n (%)* | 243 (10) |
| *50 to 80 $m^2$, n (%)* | 488 (19) |
| *80 to 100 $m^2$, n (%)* | 557 (22) |
| *100 to 120 $m^2$, n (%)* | 477 (19) |
| *≥120 $m^2$, n (%)* | 779 (31) |
| Number of cohabitants, median (IQR) | 4 (2 to 5) |
| Number of cohabitants who require care, median (IQR) | 2 (1 to 3) |
| Number of cohabitants <18 years old, median (IQR) | 2 (1 to 3) |
| Physical activity during lockdown | |
| *Not performing, n (%)* | 366 (16) |
| *Increased performing, n (%)* | 588 (25) |
| *The same performing than before lockdown, n (%)* | 492 (21) |
| *Reduced performing, n (%)* | 871 (38) |
| Alcohol consumption | |
| *Increase of alcohol consumption during lockdown, n (%)* | 111 (5) |
| *Any alcohol consumption during lockdown, n (%)* | 825 (36) |
| Any cigarette consumption during lockdown, n (%) | 229 (10) |
| Any illicit drugs consumption during lockdown, n (%) | 83 (4) |
| Any consumption of sugary drinks, n (%) | 1593 (67) |
| Concerns arising from the pandemic: degree of concern of being infected with SARS-CoV-2 | |
| *Not worried, n (%)* | 69 (3) |
| *A little worried, n (%)* | 295 (13) |
| *Moderately worried, n (%)* | 781 (33) |
| *Quite worried, n (%)* | 654 (28) |
| *Very worried, n (%)* | 533 (23) |
| Very high difficulties to cope with the job or take care of household chores, n (%) | 109 (5) |
| New health activities during lockdown, n (%) | 1187 (51) |
| Suffer any type of violence or abuse during lockdown, n (%) | 316 (14) |
| Diseases, symptoms, and medications | |

*(Continued)*

**Table 1.** (Continued)

| Variable[a] | Whole sample n = 2924 |
|---|---|
| Have or had COVID-19 | 266 (11) |
| Presence of any chronic disease | 789 (34) |
| Anxiety symptoms as measured by GAD-7 questionnaire | |
| *No anxiety (<5 points), n (%)* | 539 (23) |
| *Mild anxiety (5 to <10 points), n (%)* | 801 (35) |
| *Moderate anxiety (10 to <15 points), n (%)* | 579 (25) |
| *Severe anxiety (≥15 points), n (%)* | 384 (17) |
| *Any anxiety level (≥5 points), n (%)* | 1740 (76) |
| Depression symptoms as measured by PHQ9 questionnaire | |
| *No depression (<5 points), n (%)* | 708 (31) |
| *Mild depression (5 to <10 points), n (%)* | 700 (30) |
| *Moderate depression (10 to <15 points), n (%)* | 473 (21) |
| *Moderately severe depression (15 to <20 points), n (%)* | 256 (11) |
| *Severe depression (≥20 points), n (%)* | 161 (7) |
| *Any depression level (≥5 points), n (%)* | 1590 (69) |
| Any use of antidepressants, n (%) | 225 (10) |
| Poor or regular health self-perception | 386 (16) |

IQR = Interquartile range

GAD-7 = Generalized Anxiety Disorder Scale

PHQ9 = Patient Health Questionnaire

[a] = There were missing data (<23%) in some variables. For further details, see S1 Table.

[b] = It corresponds to the beneficiaries of the Ecuadorian Institute of Social Security (IESS, for its acronym in Spanish), the social security of the armed forces (ISSFA, for its acronym in Spanish) and the social security of the police (ISSPOL, for its acronym in Spanish). acronym in Spanish)

Regarding sociodemographic characteristics, most participants had a university education (84%) and had a full-time public or private job (63%).

Sixteen percent of the participants had regular or poor health self-perception status. In the whole sample, the prevalence of severe anxiety was 17% and severe depression was 7%; nevertheless, 76% had any anxiety level and 69% had any depression level (**Table 1**). When comparing between both sex categories (**Table 2**), moderate to severe anxiety levels (10 to ≥15 points of the GAD-7 questionnaire) were reported in 38% of the women and 29% of the men; and, moderate to severe depression levels (10 to ≥20 points of the PHQ9 questionnaire) were reported in 35% of the women and 26% of the men.

## Determinants of health self-perception

When comparing the characteristics between those participants with excellent/good *vs.* regular/poor health self-perception, there was a lower percentage of participants with a perception that the type of housing's adequacy for lockdown was poor or inadequate (12% vs. 24%, p<0.01), a lower percentage of participants with perception of greater difficulties in coping with work or taking care of household chores (3% vs. 14%, p<0.001), a lower percentage of participants with a history of COVID-19 (9% vs. 21%, p<0.001), and a lower percentage of participants with chronic diseases (28% vs. 70%, p<0.001). Anxiety and depression symptoms were less frequent among participants with good self-perception (**Table 3**).

The multivariate analyses (**Table 4**) showed that being female (aOR = 1.5, 95% CI:1.1 to 2.2), having solely public healthcare system access (aOR = 1.9, 95% CI: 1.2 to 2.9), perceiving

**Table 2. Sociodemographic characteristics, social impact variables and mental health scale of participants by sex in Ecuador during lockdown (n = 2655).**

| Variable | Women n = 1801 | Men n = 854 | p-value |
|---|---|---|---|
| Age in years of life, median (IQR) | 33 (26 to 42) | 37 (29 to 49) | 0.001 |
| Education level | | | |
| _Educational level lower than university education, n %_ | 280 (17) | 119 (15) | 0.332 |
| _University educational level or higher, n %_ | 1415 (83) | 675 (85) | |
| Employment status | | | |
| _Public or private full job, n (%)_ | 1005 (61) | 522 (67) | <0.001 |
| _Self-employment, n (%)_ | 192 (12) | 110 (14) | |
| _Unpaid work, retired or student, n (%)_ | 449 (27) | 143 (19) | |
| Access to health services | | | |
| _Social security, n (%)_ | 979 (60) | 482 (63) | 0.561 |
| _Private health insurance, n (%)_ | 396 (24) | 176 (23) | |
| _Public health services user, n (%)_ | 246 (15) | 111 (14) | |
| Perception of the adequacy of the type of housing to lockdown | | | |
| _Moderately to well adequate, n (%)_ | 1439 (86) | 679 (87) | 0.786 |
| _Little or not adequate, n (%)_ | 226 (14) | 103 (13) | |
| Housing area | | | |
| _<50 $m^2$, n (%)_ | 144 (9) | 85 (11) | 0.005 |
| _50 to 80 $m^2$, n (%)_ | 341 (17) | 130 (17) | |
| _80 to 100 $m^2$, n (%)_ | 384 (23) | 152 (20) | |
| _100 to 120 $m^2$, n (%)_ | 312 (19) | 147 (19) | |
| _≥120 $m^2$, n (%)_ | 476 (29) | 265 (34) | |
| Number of cohabitants, median (IQR) | 4 (3 to 5) | 4 (2 to 4) | 0.012 |
| Number of cohabitants who require care, median (IQR) | 2 (1 to 3) | 1 (1 to 3) | <0.001 |
| Number of cohabitants <18 years old, median (IQR) | 2 (1 to 3) | 2 (1 to 3) | 0.776 |
| Physical activity during lockdown | | | |
| _Not performing, n (%)_ | 275 (18) | 78 (11) | <0.001 |
| _Increased performing, n (%)_ | 401 (27) | 164 (23) | |
| _The same performing than before lockdown, n (%)_ | 311 (21) | 156 (22) | |
| _Reduced performing, n (%)_ | 522 (35) | 315 (44) | |
| Alcohol consumption | | | |
| _Increase of alcohol consumption during lockdown, n (%)_ | 401 (27) | 164 (23) | 0.071 |
| _Any alcohol consumption during lockdown, n (%)_ | 462 (31) | 325 (45) | <0.001 |
| Any cigarette consumption during lockdown, n (%) | 115 (8) | 103 (14) | <0.001 |
| Any illicit drugs consumption during lockdown, n (%) | 47 (3) | 36 (5) | 0.025 |
| Any consumption of sugary drinks, n (%) | 1012 (67) | 514 (72) | 0.041 |
| Concerns arising from the pandemic: degree of concern of being infected with SARS-CoV-2 | | | |
| _Not worried, n (%)_ | 35 (2) | 32 (5) | 0.003 |
| _A little worried, n (%)_ | 190 (13) | 94 (13) | |
| _Moderately worried, n (%)_ | 493 (32) | 252 (35) | |
| _Quite worried, n (%)_ | 425 (28) | 202 (28) | |
| _Very worried, n (%)_ | 376 (25) | 136 (19) | |
| Very high difficulties to cope with the job or take care of household chores, n (%) | 90 (6) | 14 (2) | <0.001 |
| New health activities during lockdown, n (%) | 779 (52) | 353 (49) | 0.336 |
| Suffer any type of violence or abuse during lockdown, n (%) | 212 (14) | 97 (14) | 0.826 |
| Diseases, symptoms, and medications | | | |
| _Have or had COVID-19_ | 172 (11) | 83 (11) | 0.746 |
| _Presence of any chronic disease_ | 512 (33) | 240 (34) | 0.873 |

_(Continued)_

**Table 2.** (Continued)

| Variable | Women n = 1801 | Men n = 854 | p-value |
|---|---|---|---|
| Anxiety symptoms as measured by GAD-7 questionnaire | | | |
| No anxiety (<5 points), n (%) | 297 (20) | 217 (30) | <0.001 |
| Mild anxiety (5 to <10 points), n (%) | 513 (34) | 244 (34) | |
| Moderate to severe anxiety (10 to ≥15 points), n (%) | 686 (38) | 250 (29) | |
| Any anxiety level (≥5 points), n (%) | 1181 (79) | 489 (69) | <0.001 |
| Depression symptoms as measured by PHQ9 questionnaire | | | |
| No depression (<5 points), n (%) | 401 (27) | 271 (38) | <0.001 |
| Mild depression (5 to <10 points), n (%) | 454 (30) | 213 (30) | |
| Moderate to severe depression (10 to ≥20 points), n (%) | 638 (35) | 226 (26) | |
| Any depression level (≥5 points), n (%) | 1092 (73) | 439 (62) | <0.001 |
| Any use of antidepressants, n (%) | 155 (10) | 65 (9) | 0.370 |
| Poor or regular health self-perception | 284 (17) | 91 (12) | 0.001 |

IQR = Interquartile range

GAD-7 = Generalized Anxiety Disorder Scale

PHQ9 = Patient Health Questionnaire

[a] = There were missing data (<23%) in some variables. For further details, see S1 Table.

[b] = It corresponds to the beneficiaries of the Ecuadorian Institute of Social Security (IESS, for its acronym in Spanish), the social security of the armed forces (ISSFA, for its acronym in Spanish) and the social security of the police (ISSPOL, for its acronym in Spanish). acronym in Spanish)

housing as inadequate to cope with lockdown (aOR = 2.2, 95% CI:1.4 to 3.4). Furthermore, perceiving very high difficulties in coping with work or managing household chores was associated with poor health self-perception (aOR = 2.7, 95% CI:1.5 to 5.0). The odds of poor self-reported health status were as high as the increase in the number of cohabitants who required care (aOR = 1.2, 95% CI:1.1 1.3). Furthermore, having had a diagnosis of COVID-19 or having had COVID-19 symptoms (aOR = 3.1, 95%CI:2.0–4.7), and suffering from chronic diseases (aOR = 6.9, 95% CI:4.9 to 9.7), having severe depression (aOR 5.9, 95%CI:3.1–11.2), were independently associated with poor health self-perception; specifically, there was a "dose-response" association between increasing depression severity and regular or poor self-perception of health; specifically, there was a 60% (65% CI:59%–83%, p-for-trend <0.001) increase in the odds of poor self-reported health status for each change to a higher depressive category.

After sensitivity analyses (S2 Table), we found similar estimates when running the final (parsimonious) model when excluding (i) high- and low-educated subjects, (ii) those with chronic diseases, (iii) those with severe anxiety, and (iv) those with severe depression.

## Differences between women and men

When comparing the variables between women and men (**Table 2**), we found that, women had significantly greater percentages of: (i) unpaid work, being retired, or being a student (27% in women vs. 19% in men, p <0.001), (ii) being extremely worried about being infected with SARS-CoV-2 (25% vs. 19%, p<0.001), (iii) extreme difficulty in coping with work or managing household chores (6% vs. 2%, p <0.001), (iv) severe anxiety (19% vs. 12%, p <0.001), (v) severe depression (8% vs. 5%, p <0.001); and (vi) poor or regular health self-perception (17% vs. 12%, p <0.001).

When we stratified the multivariate analyses of health self-perception by sex, we found differences in the determinants between men and women (see **S3 Table**). Specifically, for women, and considering the full-time job category as the reference, self-employment was

**Table 3. Differences in characteristics across health self-perception categories.**

| Variable | Good or excellent health self-perception (n = 2111) | Regular or poor health self-perception (n = 386) | p-value |
|---|---|---|---|
| Age in years of life, median (IQR) | 35 (27 to 45) | 34 (27 to 43) | 0.502 |
| Female, n % | 1350 (67) | 284 (76) | 0.001 |
| Education level | | | |
| *University educational level or higher, n %* | 1775 (86) | 303 (80) | 0.007 |
| Employment status | | | |
| *Public or private full job, n (%)* | 1296 (64) | 232 (64) | 0.006 |
| *Self-employment, n (%)* | 271 (13) | 29 (8) | |
| *Unpaid work, retired or student, n (%)* | 460 (23) | 100 (28) | |
| Access to health services | | | |
| *Social security[a], n (%)* | 1218 (61) | 221 (62) | <0.001 |
| *Private health insurance, n (%)* | 499 (25) | 61 (17) | |
| *Public health services user, n (%)* | 273 (14) | 77 (22) | |
| Perception of the adequacy of the type of housing to lockdown | | | |
| *Moderately to well adequate, n (%)* | 1860 (88) | 293 (76) | <0.001 |
| *Little or not adequate, n (%)* | 249 (12) | 92 (24) | |
| Housing area | | | |
| *<50 m2, n (%)* | 181 (9) | 56 (15) | <0.001 |
| *50 to 80 m2, n (%)* | 394 (19) | 84 (22) | |
| *80 to 100 m2, n (%)* | 445 (21) | 98 (26) | |
| *100 to 120 m2, n (%)* | 406 (19) | 57 (15) | |
| *≥120 m2, n (%)* | 674 (32) | 87 (23) | |
| Number of cohabitants, median (IQR) | 4 (2 to 5) | 4 (3 to 5) | 0.039 |
| Number of cohabitants who require care, median (IQR) | 2 (1 to 3) | 2 (1 to 3) | <0.001 |
| Number of cohabitants <18 years old, median (IQR) | 2 (1 to 3) | 2 (1 to 3) | <0.084 |
| Physical activity during lockdown | | | |
| *Not performing, n (%)* | 274 (14) | 87 (25) | <0.001 |
| *Increased performing, n (%)* | 521 (27) | 67 (19) | |
| *The same performing than before lockdown, n (%)* | 433 (22) | 57 (16) | |
| *Reduced performing, n (%)* | 730 (37) | 136 (39) | |
| Alcohol consumption | | | |
| *Increase of alcohol consumption during lockdown, n (%)* | 88 (5) | 22 (6) | 0.136 |
| *Any alcohol consumption during lockdown, n (%)* | 724 (37) | 98 (28) | 0.002 |
| Any cigarette consumption during lockdown, n (%) | 193 (10) | 36 (10) | 0.734 |
| Any illicit drugs consumption during lockdown, n (%) | 68 (3) | 15 (4) | 0.427 |
| Any consumption of sugary drinks, n (%) | 1338 (68) | 246 (71) | 0.369 |
| Concerns arising from the pandemic: degree of concern of being infected with SARS-CoV-2 | | | |
| *Not worried, n (%)* | 60 (3) | 9 (3) | <0.001 |
| *A little worried, n (%)* | 250 (13) | 45 (13) | |
| *Moderately worried, n (%)* | 703 (36) | 75 (21) | |
| *Quite worried, n (%)* | 533 (27) | 118 (34) | |
| *Very worried, n (%)* | 425 (22) | 103 (29) | |
| Very high difficulties to cope with the job or take care of household chores, n (%) | 57 (3) | 50 (14) | <0.001 |
| New healthy or socially active activities during lockdown, n (%) | 1039 (53) | 143 (41) | |
| Suffer any type of violence or abuse during lockdown, n (%) | 224 (12) | 80 (23) | <0.001 |
| Diseases, symptoms, and medications | | | |

*(Continued)*

**Table 3.** (Continued)

| Variable | Good or excellent health self-perception (n = 2111) | Regular or poor health self-perception (n = 386) | p-value |
|---|---|---|---|
| Have or had COVID-19 | 185 (9) | 79 (21) | <0.001 |
| Presence of any chronic disease | 551 (28) | 237 (70) | <0.001 |
| Anxiety symptoms as measured by GAD questionnaire, median (IQR) | 7 (4 to 12) | 13 (8 to 17) | <0.001 |
| _No anxiety (<5 points), n (%)_ | 504 (26) | 32 (9) | <0.001 |
| _Mild anxiety (5 to <10 points), n (%)_ | 727 (37) | 73 (21) | |
| _Moderate anxiety (10 to <15 points), n (%)_ | 464 (24) | 112 (33) | |
| _Severe anxiety (≥15 points), n (%)_ | 255 (13) | 125 (37) | |
| _Any anxiety level (≥5 points), n (%)_ | 1429 (73) | 903 (89) | <0.001 |
| Depression symptoms as measured by PHQ9 questionnaire, median (IQR) | 7 (3 to 11) | 13 (8 to 18) | <0.001 |
| _No depression (<5 points), n (%)_ | 662 (34) | 43 (13) | <0.001 |
| _Mild depression (5 to <10 points), n (%)_ | 625 (32) | 74 (22) | |
| _Moderate depression (10 to <15 points), n (%)_ | 385 (20) | 87 (26) | |
| _Moderately severe depression (15 to <20 points), n (%)_ | 180 (9) | 74 (22) | |
| _Severe depression (≥20 points), n (%)_ | 94 (5) | 63 (18) | |
| _Any depression level (≥5 points), n (%)_ | 1284 (66) | 298 (88) | <0.001 |
| Any use of antidepressants, n (%) | 157 (8) | 65 (19) | <0.001 |

IQR = Interquartile range

GAD-7 = Generalized Anxiety Disorder Scale

PHQ9 = Patient Health Questionnaire

[a] = It corresponds to the beneficiaries of the Ecuadorian Institute of Social Security (IESS, for its acronym in Spanish), the social security of the armed forces (ISSFA, for its acronym in Spanish) and the social security of the police (ISSPOL, for its acronym in Spanish). acronym in Spanish)

significantly associated with a poorer self-reported health status (aOR = 0.6, 95%CI:0.2 to 0.8); moreover, there were 30% higher odds of poor self-reported health status per each extra cohabitant who required care (aOR = 1.3, 95% CI:1.1 to 1.5). Those having or having had COVID-19 had 330% higher odds of regular or poor self-reported health status (aOR = 4.3, 95% CI:2.6 to 7.1) when compared with their counterparts; and those reporting any chronic disease had a greater likelihood of having poor self-reported health status (aOR = 8.1, 95% CI:5.3 to 12.1). We did not find such strong associations in men. For men, by contrast, we found that those perceiving housing's adequacy for lockdown as poor or inadequate had greater odds of regular or poor self-reported status than women (aOR = 3.2, 95% CI:1.4 to 7.1), and depression was much more associated with regular or poor self-reported health status than for women (aOR = 1.70, 95% CI:1.32 to 2.22).

## Discussion

Our main findings were that the determinants of regular or poor self-reported health status among Ecuadorian-surveyed adult persons were: *(i)* being female by comparison with being male, *(ii)* perceiving housing conditions as inadequate for coping with the lockdown, *(iii)* living with people who require care, *(v)* perceiving extreme difficulties in coping with both work exigencies or managing household chores, *(vi)* a history of COVID-19 infection, *(vii)* presence of chronic diseases, and *(viii)* depressive symptoms. Similar to other studies, the complex Ecuadorian context during lockdown [10, 13–16, 18] helps explain the accentuated impact of such factors on the self-reported health status of the Ecuadorian population.

**Table 4. Crude and adjusted odds ratios of regular or bad health self-perception.**

| Variable | Crude | p-value | Saturated | p-value | Parsimonious | p-value |
|---|---|---|---|---|---|---|
| Female *(male is the ref.)* | 1.5 (1.2 to 2.0) | 0.001 | 1.7 (1.1 to 2.5) | 0.013 | 1.5 (1.1 to 2.2) | 0.023 |
| Education level | | | | | | |
| *University educational level or higher (lower than university education is the ref.)* | 0.7 (0.5 to 0.9) | 0.007 | 0.8 (0.5 to 1.3) | 0.351 | - | - |
| Employment status | | | | | | |
| *Public or private full job (ref.)* | 1 | - | 1 | - | 1 | - |
| *Self-employment* | 0.6 (0.4 to 0.9) | 0.013 | 0.4 (0.2 to 0.8) | 0.008 | 0.5 (0.3 to 0.8) | 0.011 |
| *Unpaid work, retired or student* | 1.2 (0.9 to 1.6) | 0.139 | 0.9 (0.6 to 1.4) | 0.574 | 0.8 (0.5 to 1.2) | 0.359 |
| Access to health services | | | | | | |
| *Social security[a] (ref.)* | 1 | - | 1 | - | 1 | - |
| *Private health insurance* | 0.7 (0.5 to 0.9) | 0.010 | 0.7 (0.4 to 1.1) | 0.134 | 0.7 (0.5 to 1.1) | 0.162 |
| *Public health services user* | 1.6 (1.2 to 2.1) | 0.003 | 1.8 (11 to 2.9) | 0.022 | 1.9 (1.2 to 2.9) | 0.009 |
| Perception of the adequacy of the type of housing to lockdown | | | | | | |
| *Little or not adequate (Moderately to well adequate is ref.)* | 2.4 (1.8 to 3.1) | <0.001 | 2.1 (1.3 to 3.3) | 0.003 | 2.2 (1.4 to 3.4) | <0.001 |
| Housing area | | | | | | |
| *<50 m2 (ref)* | 1 | - | 1 | - | - | - |
| *50 to 80 m2* | 0.7 (0.5 to 1.0) | 0.056 | 0.6 (0.3 to 1.3) | 0.215 | - | - |
| *80 to 100 m2* | 0.7 (0.5 to 1.0) | 0.073 | 0.7 (0.4 to 1.5) | 0.363 | - | - |
| *100 to 120 m2* | 0.5 (0.3 to 0.7) | <0.001 | 0.6 (0.3 to 1.2) | 0.174 | - | - |
| *≥120 m2* | 0.4 (0.2 to 0.4) | <0.001 | 0.5 (0.3 to 1.1) | 0.096 | - | - |
| Number of cohabitants who require care *(per each increase in one cohabitant)* | 1.3 (1.2 to 1.4) | <0.001 | 1.2 (1.1 to 1.4) | 0.002 | 1.2 (1.1 to 1.3) | 0.004 |
| Physical activity during lockdown | | | | | | |
| *Increased performing (any increase or no performing is ref.)* | 0.7 (0.5 to 0.9) | 0.004 | 0.9 (0.6 to 1.5) | 0.793 | - | - |
| Alcohol consumption | | | | | | |
| *Increase of alcohol consumption during lockdown, n (%)* | 1.4 (0.9 to 2.3) | 0.138 | 1.3 (0.6 to 2.6) | 0.525 | - | - |
| Concerns arising from the pandemic: degree of concern of being infected with SARS-CoV-2 | | | | | | |
| *Nothing worried (ref)* | 1 | - | 1 | - | 1 | - |
| *A little worried* | 1.2 (0.6 to 2.6) | 0.642 | 0.6 (0.2 to 1.7) | 0.331 | - | - |
| *Moderately worried, n (%)* | 0.7 (0.3 to 1.5) | 0.367 | 0.8 (0.3 to 2.3) | 0.697 | - | - |
| *Quite worried, n (%)* | 1.5 (0.7 to 3.1) | 0.295 | 1.2 (0.4 to 3.4) | 0.680 | - | - |
| *Very worried, n (%)* | 1.7 (0.8 to 3.4) | 0.200 | 0.8 (0.3 to 2.3) | 0.680 | - | - |
| Very high difficulties to cope with the job or take care of household chores *(not having is the ref.)* | 5.6 (3.7 to 8.3) | <0.001 | 2.6 (1.4 to 5.0) | 0.004 | 2.7 (1.5 to 5.0) | 0.002 |
| New healthy or socially active activities during lockdown *(not having is the ref.)* | 0.6 (0.5 to 0.8) | <0.001 | 0.7 (0.5 to 1.1) | 0.109 | - | - |
| Suffer any type of violence or abuse during lockdown *(not having is the ref.)* | 2.22 (1.7 to 3.0) | <0.001 | 1.29 (0.8 to 2.0) | 0.247 | - | - |
| Diseases, symptoms, and medications | | | | | | |
| *Have or had COVID-19 (not having is the ref.)* | 2.7 (2.0 to 3.6) | <0.001 | 3.1 (2.0 to 4.9) | <0.001 | 3.1 (2.0 to 4.7) | <0.001 |
| *Presence of any chronic disease (not having is the ref.)* | 6.1 (4.7 to 7.8) | <0.001 | 6.9 (4.8 to 10.0) | <0.001 | 6.9 (4.9 to 9.7) | <0.001 |
| Anxiety symptoms as measured by GAD-7 questionnaire | | | | | | |
| *No anxiety (<5 points) (ref.)* | 1 | - | 1 | - | - | - |
| *Mild anxiety (5 to <10 points)* | 1.6 (1.0 to 2.4) | 0.037 | 1.1 (0.6 to 2.1) | 0.723 | - | - |
| *Moderate anxiety (10 to <15 points)* | 3.8 (2.5 to 5.7) | <0.001 | 1.2 (0.6 to 2.5) | 0.545 | - | - |
| *Severe anxiety (≥15 points)* | 7.7 (5.1 to 11.7) | <0.001 | 1.2 (0.6 to 2.7) | 0.631 | - | - |
| Depression symptoms as measured by PHQ-9 questionnaire, median (IQR) | | | | | | |
| *No depression (<5 points) (ref.)* | 1 | - | 1 | - | 1 | - |

*(Continued)*

**Table 4.** (Continued)

| Variable | Crude | p-value | Saturated | p-value | Parsimonious | p-value |
|---|---|---|---|---|---|---|
| *Mild depression (5 to <10 points)* | 1.8 (1.2 to 2.7) | 0.003 | 1.1 (0.6 to 2.1) | 0.666 | 1.4 (0.9 to 2.3) | 0.162 |
| *Moderate depression (10 to <15 points)* | 3.5 (2.4 to 5.1) | <0.001 | 2.5 (1.3 to 4.6) | 0.005 | 3.2 (2.0 to 5.3 | <0.001 |
| *Moderately severe depression (15 to <20 points)* | 6.3 (4.2 to 9.5) | <0.001 | 2.7 (1.3 to 5.5) | 0.008 | 3.8 (2.2 to 6.7) | <0.001 |
| *Severe depression (≥20 points)* | 10.3 (6.6 to 16.0) | <0.001 | 4.3 (1.8 to 10.0) | 0.001 | 5.9 (3.1 to 11.2) | <0.001 |
| *p-for-trend* | | | | | 1.6 (1.39 to 1.8) | <0.001 |
| Any use of antidepressants *(not using is the ref.)* | 2.6 (1.9 to 3.6) | <0.001 | 1.3 (0.8 to 2.2) | 0.270 | - | - |

GAD-7 = Generalized Anxiety Disorder Scale

PHQ9 = Patient Health Questionnaire

[a] = It corresponds to the beneficiaries of the Ecuadorian Institute of Social Security (IESS, for its acronym in Spanish), the social security of the armed forces (ISSFA, for its acronym in Spanish) and the social security of the police (ISSPOL, for its acronym in Spanish). acronym in Spanish)

The factors associated with self-reported health differed by sex/gender. Specifically, women had a greater negative impact on their self-reported health when they received only public health services, had inadequate housing for lockdown, an increasing number of cohabitants who required care in the family, extreme perceived difficulties coping with work or managing household chores, having COVID-19, the presence of chronic disease, and increasing depressive symptoms. In men, the determinants were inadequate housing type, presence of chronic disease, and increasing depressive symptoms. Interestingly, in women, COVID-19's effect and the presence of chronic diseases were more accentuated than in men as in other studies have been found [43]; while in men, the housing inadequacy and increasing depressive symptoms were more accentuated than in women, as was found in other contexts such as Spain [27]. We could not demonstrate a significant interaction or effect modification by sex/gender on the association between the associated factors and self-reported health.

Furthermore, the percentage of poor or regular health self-perception was 16%, which was greater among women (17%) than among men (12%). The prevalence of poor self-reported health status (16%) is greater than that found in other contexts during lockdown [40]. A comparison between our data and the results of an occupational health survey conducted among workers in Quito and Guayaquil before the pandemic, where the authors found a prevalence of 11% for self-perceived poor health status [36], suggests that the lockdown has significantly worsened the levels of self-perceived health in Ecuador. This is possibly because of the weakened social protection system, reflected on the lack of social support policies during this pandemic phase. Given that safe and adequate housing is essential to protect people from environmental conditions, create social ties, and establish life projects, housing deprivation and the lack of an adequate urban environment have significant health consequences for both sexes [44].

These findings highlight the impact of gender inequality on the burden of care and domestic work, and its negative effect on women's health. As other researchers have shown [7, 8], being a woman was a risk factor for increased mental distress during lockdown. Furthermore, our findings add data on unpaid care and domestic work as one of the mechanisms through which lockdowns affect women's health in particular ways [45]. Men did not feel this impact, which is probably related to the assignment of traditional gender roles, in which men are usually not responsible for care and domestic work.

Although it is true female had a poorer perception of self-reported health; both, men and women suffered a significant impact on their health, especially in the area of mental health.

The fact that we found a worse self-perceived health in women compared to men is coincident with the results obtained from previous studies [46, 47].

Although mortality from COVID-19 has been consistently higher in men [48], scientific literature shows that women tend to have worse living conditions and use health services more frequently; they also have a greater number of disease diagnoses. A possible explanation is that women have historically suffered the greatest burden of social inequalities and have assumed more home responsibilities, which, compounded by the added responsibilities during confinement [49], has entailed a greater stress load. In addition, our findings corroborate that women have a greater burden of caring for dependent people, a fact explained by the assignment of traditional gender roles that persist in patriarchal systems, such as the Ecuadorian society. At the same time, these results show the poorness of the social protection system to deal with care as a social risk managed by family ties and networks, instead of public services. Both forms of response were eliminated during lockdown, so that care fell to women from nuclear families.

Importantly, women suffer more from the lack of proper access to tele-education for their children, given the cultural tendency to assign women as those responsible for educating children, despite their having to cope with everyday duties such as productive work. The regional difficulties in educating children could explain the even more important health effect of caring for household children. Specifically, in Latin America and the Caribbean, schools have been closed for an average of 37 weeks since March 2020. In Ecuador, it has been 40 weeks. In addition, only 39% of primary school students can read a simple text [50], and only 37% of households have Internet access, which means that six out of ten children cannot study through digital platforms. The situation is more serious for children in rural areas, where only 16% of households have this service [51].

Males were affected too; as in other studies' findings, there is poorer self-reported health in men when inadequate housing conditions are perceived [52, 53], and when there are depressive symptomatology [54]. These findings could be explained by: First, the fact that men are more prone to discomfort because of housing conditions, which is in turn related to gender roles and its impact on self-reported health. In traditional patriarchal systems, men are seen as the primary family breadwinners, and not being able to meet a desired standard could affect men's health more than women's. Second, according to previous studies [54], potential explanations of the accentuation of poor self-reported health by depression are related with the fact that the presence of symptoms of depression have been associated with the occurrence of severe sexual functioning disorders; and third, the intensity of other fears, not measured in this survey, could be correlated with depressive and sexual disorders.

As a result of overcrowding, couples experience financial and family stressors, with an increase in the number of conflicts during sustained social isolation and physical proximity, particularly among young and newly-formed intimate relationships. Moreover, with housing insecurity and housing conditions [55] inadequate to tackle the lockdown, a poor self-rated health status is expected. In that sense, we corroborated that living in poor housing conditions and having low income and/or poor labor conditions–the social determinants of health–affect more vulnerable populations [27, 56].

In addition, we found that 38% of women and 29% of men reported moderate to severe anxiety, and moderate to severe depression levels were reported in 35% of women and 26% of men. We corroborated that there were gender differences in depression and anxiety, as well as differences in the quantity of (subclinical) depressive symptoms [19]. Prior studies found that the levels of such mental health problems were much lower than our findings. Specifically, in one study in Spain [27], 31% of women and 18% of men reported moderate to severe anxiety, and moderate to severe depression levels were reported in 29% of women and 17% of men. It is possible that specific contexts determine reactions to the lockdown. We believe that the

Ecuadorian population felt a profound health impact owing to the poor and uncoordinated pandemic response from national and local authorities.

Moreover, the frequency of anxiety and depression levels was greater in females than in males in the study population. However, the association between depression symptoms and regular-to-poor health self-perception was stronger in men than in women. It is coincident with a study [54] that found that the fear of contracting the COVID-19 infection, the fear of the health condition of loved ones, depressed mood, and exposition to media reports worsened their mental health. Furthermore, another study found that psychosocial factors explained the highest proportion of the variance in anxiety symptoms, being even higher in men than in women. Also, we found that depression and adverse housing conditions were more associated to a poorer self-perceived health in men than in women, maybe explained by previous findings that depressive symptomatology has a greater impact on men's health when compared to women in terms of suicide attempts [57]. While suicide attempt rates are similar between men and women, males have an almost threefold higher risk of dying by suicide than females [58]. This higher mortality among men could be explained by various factors, including the use of more lethal means (firearms and hanging methods), whereas drug intoxication is more frequent in women. Young men may be less predisposed to help-seeking behaviors as an attempt to exhibit masculine behaviors, and their tendency to adopt avoidance strategies may make it more difficult for them to cope with emotional and behavioral problems [49]. In that regard, specific pandemic and lockdown policies should use a gender approach to identify those at risk and intervene [27].

Regarding protective variables, perceived social support was independently associated with lower anxiety while intimate partner violence was further associated with higher anxiety symptoms and this pattern was consistent in men and women.

Serious questioning of the Ecuadorian Public Health System and its manner of operating was a serious pandemic effect. Some participants reported being part of or knowing someone who was close to the health system. The lack of response to emergencies (which involved life or death in many cases), lack of basic information regarding COVID-19, serious difficulties in communicating basic information to the population, and lack of psychological support spaces for front-line professionals were the most common problems. Through proposals from civil society and academia the population gained access to spaces for psychological support and crisis intervention [59]. The evidence on the lockdown's gendered impact on self-perceived health found here is a strong indicator of how deep-rooted patriarchal gender beliefs affect the health of the Ecuadorian population. Even though we have found only one study in Ecuador on teleworking's impact on people's lives, especially women, there are clear impacts from increases in teleworker numbers in the sectors where it was implemented, in the codification of work schedules and conciliation with family life, as well as significant specific effects on unions and teleworking health and safety [20].

The pandemic's impact is also unknown in areas related to the additional burdens and impacts resulting from teleworking on women, who often bear the bulk of household care work; protection of labor rights and workers in legislative frameworks; and judgments or constitutional revisions relating to teleworking laws. Specifically, in a study it was found that 76% of the women surveyed indicate an increase in workload, and 56% dedicate themselves to their children's schoolwork [20]. Furthermore, women who spent long hours on housework and childcare were more likely to report increased levels of psychological distress, and women were more likely than men to reduce working hours and adapt employment schedules because of increased unpaid care time [60]. In that regard, our results can be used to better justify formulating regulations that guarantee a maximum of eight daily working hours [20]. Continued gender inequality in divisions of unpaid care work during lockdown may put women at a

greater risk of psychological distress, which is a consequence of gender biases in divisions of labor and their impact on psychological health [60]. Teleworking also raised questions about the use of the physical household spaces, which in many cases implied no temporal limits between hours of outside and inside home labor. In these cases, women mostly assume these new daily dynamics, resulting in extreme fatigue, anxiety, and sadness.

The COVID-19 prevalence was around 11%, which is similar to published (and non-published) reports of the pandemic's evolution in Ecuador during the surveyed months [61]. It is important to enhance this finding because it helps understanding of this population's context during the specific survey period.

This study has several limitations. The survey was only available online and was mainly completed by highly educated respondents, and it may have excluded people without digital access. As it happened with many studies during lockdown, an electronic survey had to be applied to obtain information in that setting. The advantage of this approach was that relevant information could be obtained in a time were performing research had many challenges. The disadvantage was that access to the survey could be limited to some populations (higher socioeconomic status, younger age, etc.). Thus, we speculate that the associations and inequalities would be even greater if it included more vulnerable people. Non-representative responses are a handicap of online surveys, since they do not capture the responses of those who lack access and/or Internet skills (e.g., the elderly, those with lower education, or those in remote locations). In addition, we had more responses from women than from men. This requires strategies to ensure greater male participation. However, as the analyses were stratified by sex, the main results were compared with those of the reference group. We plan to conduct another survey and a qualitative study several months after this first survey. This will be carried out using the same method, as it is being applied in other Latin American countries and Spain, to compare the results in different populations.

## Conclusions

Being female, having only access to the public healthcare system, having a perception of inadequate housing, living with cohabitants who require care, perceiving very high difficulties in coping with work or managing household chores, having or having had COVID-19, the presence of chronic diseases, and depressive symptoms are associated with a poorer self-reported health status in Ecuadorian population. Conversely, self-employment and having private health insurance were significantly and independently associated with better self-reported health status. For women, self-employment, having solely public healthcare system access, perceiving housing conditions as inadequate, having cohabitants requiring care, having very high difficulties to cope with household chores, having COVID-19, and having a chronic disease increased the likelihood of having poor self-reported health status. For men, poor or inadequate housing, presence of any chronic disease, and depression increased the likelihood of having poor self-reported health status.

## Supporting information

**S1 Data.**
(CSV)

**S1 File. Spanish version of the survey.**
(DOCX)

**S1 Table. Description of the sample and missing values per each variable.**
(DOCX)

**S2 Table. Adjusted odds ratios of regular or bad health self-perception excluding: (i) high and low educated subjects, (ii) those with chronic diseases, (iii) those with severe anxiety, and (iv) those with severe depression.**
(DOCX)

**S3 Table. Adjusted odds ratios of regular or bad health self-perception stratified by sex according to the parsimonious logistic regression model of Table 3 of the main text.**
(DOCX)

## Acknowledgments

We thank the project members for their input and the *Instituto de Salud Pública* from *Pontificia Universidad Católica del Ecuador* for their contributions. We also thank Anna Moleras, who prepared the database, and the study participants.

## Author Contributions

**Conceptualization:** Iván Dueñas-Espín, Constanza Jacques-Aviñó, Andrés Peralta.

**Data curation:** Iván Dueñas-Espín, Constanza Jacques-Aviñó, Andrés Peralta.

**Formal analysis:** Iván Dueñas-Espín.

**Investigation:** Iván Dueñas-Espín, Constanza Jacques-Aviñó, Verónica Egas-Reyes, Sara Larrea, Ana Lucía Torres-Castillo, Patricio Trujillo, Andrés Peralta.

**Methodology:** Iván Dueñas-Espín, Andrés Peralta.

**Validation:** Iván Dueñas-Espín, Sara Larrea, Andrés Peralta.

**Writing – original draft:** Iván Dueñas-Espín, Constanza Jacques-Aviñó, Sara Larrea, Andrés Peralta.

**Writing – review & editing:** Iván Dueñas-Espín, Constanza Jacques-Aviñó, Verónica Egas-Reyes, Sara Larrea, Ana Lucía Torres-Castillo, Patricio Trujillo, Andrés Peralta.

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
