## [Decision Letter · Decision Letter 0]

6 Nov 2022

PONE-D-22-26130Determinants of self-reported health status during COVID-19 lockdown among surveyed Ecuadorian population: a cross sectional studyPLOS ONE

Dear Dr. Dueñas-Espín,

Thank you for submitting your manuscript to PLOS ONE. After careful consideration, we feel that it has merit but does not fully meet PLOS ONE’s publication criteria as it currently stands. Therefore, we invite you to submit a revised version of the manuscript that addresses the points raised during the review process.

We look forward to receiving your revised manuscript.

Kind regards,

Hadi Ghasemi

Academic Editor

PLOS ONE

Journal Requirements:

Reviewers' comments:

Reviewer's Responses to Questions

**Comments to the Author**

1. Is the manuscript technically sound, and do the data support the conclusions?

Reviewer #1: Partly

Reviewer #2: Yes

Reviewer #3: Partly

2. Has the statistical analysis been performed appropriately and rigorously? 

Reviewer #1: Yes

Reviewer #2: Yes

Reviewer #3: I Don't Know

3. Have the authors made all data underlying the findings in their manuscript fully available?

Reviewer #1: Yes

Reviewer #2: Yes

Reviewer #3: Yes

4. Is the manuscript presented in an intelligible fashion and written in standard English?

Reviewer #1: Yes

Reviewer #2: Yes

Reviewer #3: Yes

5. Review Comments to the Author

Reviewer #1: The study lacks originality and similar research has been published which provides substantial information in this area of research. examples include:

Hidalgo-Andrade P, Hermosa-Bosano C, Paz C. Teachers’ mental health and self-reported coping strategies during the COVID-19 pandemic in Ecuador: A mixed-methods study. Psychology research and behavior management. 2021;14:933.

Bates BR, Moncayo AL, Costales JA, Herrera-Cespedes CA, Grijalva MJ. Knowledge, attitudes, and practices towards COVID-19 among Ecuadorians during the outbreak: an online cross-sectional survey. Journal of Community Health. 2020 Dec;45(6):1158-67.

The article fails to add to the existing knowledge base in this area of research.

Reviewer #2: The study titled "Determinants of self-reported health status during COVID-19 lockdown among surveyed Ecuadorian population: a cross sectional study" is written very well. I just have two major questions, that needs clarification to enhance the manuscript.

First, the study conducted backed in July 2020 to October 2020, mean two years back, they why the findings are reported so late, there should an explanation for this in the manuscript.

Second, the authors stated that "Their median (IQR) age was 34 (27–44) years, most 33 participants had a university education (84%) and a full-time public or private job (63%); 34 16% of participants had poor health self-perception". This clearly indicated that there is selection bias in the study participants. For example, in a country as per the manuscript " In the same year, 24.2% of the urban population and 49.2% of the rural population lived under the poverty line [16]. Moreover, only approximately 30% of the economically active population had an adequate job...", where the genral population is 30% economically active, and the study sample size is 63% full-time job, this is clearly selction-bias. So the authors should take into consideration this poin and revised the mansucripts mention this part in limittaions or methods. As per this comment the whole analysis and discussion part needs to be revised.

Reviewer #3: The study duration in the abstract mentions July to October 2020, while the introduction states that it was conducted between March to October 2020

The article states “We conducted a cross-sectional study of adults living in Ecuador between July and October 2020”. Does this imply the duration of the study or that the adults were in Ecuador during this time period? This point needs to be made clear in the abstract and introduction

Table 1 and Table 2 have missing data. For example, Table 1 mentions there are 1801 females but does not mention males. In addition, the Education level section mentions that 408 individuals do not have university level education while 2191 individuals do. The total number (2191+408) does not equal to 2924 which is the total sample size. This mistake is evident throughout Table 1 and Table 2.

The paragraph below Table 1 mentions moderate anxiety levels to be 76% while the Table 1 states that it is 25%

The paragraph below Table states “Severe anxiety and depression were present in 17% of women and 12% of men.” Where are these percentages taken from as Table 2 includes anxiety and depression as two different variables with different percentages between males and females. How were the percentages of 17 and 12 calculated?

Table 1 and Table 2 have rounded percentages. However, the paragraph below Table 1 includes percentages such as 38.1%, 29.3%, 35.4%,26.5%. These percentages are not mentioned in Table 1 or Table 2.

The article seeks to demonstrate that males and females are both affected with females being more affected. The authors then discuss the different determinants affecting both males and females. However, the discussion section only mentions “being female” as a determinant of “poor self-reported health status”.

The conclusion section also makes specific mention of “being female” as a determinant and goes on to ignore males who were discussed in depth in the preceding section.

6. PLOS authors have the option to publish the peer review history of their article (what does this mean?). If published, this will include your full peer review and any attached files.

Reviewer #1: No

Reviewer #2: **Yes: **Junaid Ahmad

Reviewer #3: No

---

## [Author Response · Author response to Decision Letter 0]

26 Jan 2023

Rebuttal letter to reviewers’ comments to authors

PLOS ONE 

Manuscript ID: PONE-D-22-26130

Title: Determinants of self-reported health status during COVID-19 lockdown among surveyed Ecuadorian population: a cross sectional study

Iván Dueñas-Espín, Constanza Jacques- Aviñó, Verónica Egas-Reyes, Sara Larrea, Ana Lucía Torres-Castillo, Patricio Trujillo, Andrés Peralta

Decision: Revision required

Article Type: original article.

Corresponding Author: Iván Dueñas-Espín

We thank the Editor and Reviewers for their insightful comments that have been helpful to improve the manuscript. These comments are very much appreciated. While it is known that COVID-19 lockdown puts people at risk of stress, anxiety depression and poor self-reported health, we found that being female, having solely public healthcare and social protection systems access, perceiving housing conditions as inadequate, living with cohabitants requiring care, perceiving difficulties in coping with work or managing household chores, COVID-19 infection, chronic disease, and having had depression symptoms before the pandemic were significantly and independently associated with poor self-reported health status in Ecuadorian population. Furthermore, our findings highlight the impact of gender inequality on the burden of care and domestic work, and its negative effect, especially, on women’s health. This is probably related to the assignment of traditional gender roles, in which men are usually not responsible for care and domestic work, and principally, the loss of care networks that characterized the lockdowns. 

We hope that our answers satisfy the questions and comments from the Editorial team and the reviewers.

 

JOURNAL REQUIREMENTS:

ANSWER:

We have reviewed the guidelines and changed the manuscript accordingly. Please see marked and unmarked copies of the manuscript. 

ANSWER:

Thank you very much. After a brief discussion with the authors of the manuscript we decided to make the data public for proper replication of our results. We have changed the “Data Availability” statement in this way: “All relevant data are within the manuscript and its Supporting Information files”. Therefore, we share the database as a supporting information file “S1 Data” as a “.csv” file. In that regard, we have changed the Data Availability statement as follows: 

“Data Availability Statement: All relevant data are within the manuscript and its Supporting information files.”

This new statement has been mentioned in the new covering letter.

ANSWER:

Thank you very much. We have included captions in the “Supporting Information” files as guidelines states.

COMMENTS TO THE AUTHOR

1. Is the manuscript technically sound, and do the data support the conclusions?

Reviewer #1: Partly

Reviewer #2: Yes

Reviewer #3: Partly

ANSWER:

Thank you very much.

2. Has the statistical analysis been performed appropriately and rigorously?

Reviewer #1: Yes

Reviewer #2: Yes

Reviewer #3: I Don't Know

ANSWER:

Thank you very much.

3. Have the authors made all data underlying the findings in their manuscript fully available?

The PLOS Data policy requires authors to make all data underlying the findings described in their manuscript fully available without restriction, with rare exception (please refer to the Data Availability Statement in the manuscript PDF file). The data should be provided as part of the manuscript or its supporting information or deposited to a public repository. For example, in addition to summary statistics, the data points behind means, medians and variance measures should be available. If there are restrictions on publicly sharing data—e.g. participant privacy or use of data from a third party—those must be specified.

Reviewer #1: Yes

Reviewer #2: Yes

Reviewer #3: Yes

ANSWER:

Thank you very much.

4. Is the manuscript presented in an intelligible fashion and written in standard English?

Reviewer #1: Yes

Reviewer #2: Yes

Reviewer #3: Yes

ANSWER:

Thank you very much.

 

5. Review Comments to the Author

ANSWERS TO REVIEWERS

ANSWERS TO REVIEWER #1:

Reviewer #1: The study lacks originality and similar research has been published which provides substantial information in this area of research. examples include:

Hidalgo-Andrade P, Hermosa-Bosano C, Paz C. Teachers’ mental health and self-reported coping strategies during the COVID-19 pandemic in Ecuador: A mixed-methods study. Psychology research and behavior management. 2021;14:933.

Bates BR, Moncayo AL, Costales JA, Herrera-Cespedes CA, Grijalva MJ. Knowledge, attitudes, and practices towards COVID-19 among Ecuadorians during the outbreak: an online cross-sectional survey. Journal of Community Health. 2020 Dec;45(6):1158-67.

The article fails to add to the existing knowledge base in this area of research.

ANSWER:

Thanks a lot for your important comment. We have reviewed the last publications on the topic of social impact of COVID-19 lockdown in Ecuador and found several papers. We also found that those previous studies did not evaluate the determinants of mental or self-reported health during lockdown in a comprehensive way, nor did most of them investigate the differences by gender/sex. Therefore, despite that several papers have been published, our study contributes with fresh data explaining how the determinants of self-reported and mental health affected the general population during lockdown; and, specifically, how the lockdown circumstances impacted men and women differentially. In addition, our study provides valuable information to identify the population at risk of poor results in mental health and self-reported health; and it provides with elements to develop future interventions, stratified by gender, that help to tackle social impacts from lockdowns and to maintain the general well-being and mental health of the population. Also, we believe that these findings are clear hints of the weakening of health and social protection systems, which leave families, especially women, to bear and manage social risk exacerbated during lockdown. 

To incorporate the reviewer's comments, we added the following paragraphs in the “Introduction” section, lines 110 to 135, pages 5 and 6:

“After a brief revision from literature in Medline, we found several papers studying the social impact from the COVID-19 lockdown in Ecuadorian population. Regarding its impact on lifestyles, one study (1) found that teachers were not ready for the sudden shift to emergency remote teaching. Another study (4) found that stress was associated with poorer diet quality. Therefore, the confinement affected various areas of the lives of citizens.

Regarding the impact of lockdown on mental health of Ecuadorian population, one study (2) found that burnout has a mediating effect between job motivation and turnover intention, and that female and male workers’ burnout and turnover intentions levels are different when intrinsic motivation is present. Otherwise, a multicenter study (3) showed that the higher perception of stress, the less self-care activities are adopted, and in turn the lower the beneficial effects on well-being. 

Regarding knowledge, attitudes and practices towards COVID-19, a paper (6) found that participants reported high levels of adoption of preventive practices; importantly, unemployed individuals, househusbands/housewives, or manual laborers, as well as those with an elementary school education, have lower levels of knowledge about COVID-19.

In the mental health area, a paper (5) found that cognitive emotion regulation strategies on anxiety and depression was moderated by the sex of participants and the time of assessment. Moreover, a study (7) found that age was significantly correlated with all the psychological variables; importantly, females presented higher levels of stress, especially those who have home care responsibilities. 

Thus, despite that several papers have been published, it is not clear how the determinants of self-reported and mental health affected the general population during lockdown; and, specifically, how the lockdown circumstances affected to men and women differentially.”

References:

1. Estrella F. Ecuadorian university English teachers’ reflections on emergency remote teaching during the COVID-19 pandemic. Int J Educ Res Open. 2022;3(January). 

2. Paredes-Aguirre MI, Barriga Medina HR, Campoverde Aguirre RE, Melo Vargas ER, Armijos Yambay MB. Job Motivation, Burnout and Turnover Intention during the COVID-19 Pandemic: Are There Differences between Female and Male Workers? Healthc. 2022;10(9). 

3. Luis E, Bermejo-Martins E, Martinez M, Sarrionandia A, Cortes C, Oliveros EY, et al. Relationship between self-care activities, stress and well-being during COVID-19 lockdown: A cross-cultural mediation model. BMJ Open. 2021;11(12). 

4. Abril-Ulloa V, Santos SPL dos, Morejón-Terán YA, Carpio-Arias TV, Espinoza-Fajardo AC, Vinueza-Veloz MF. Stress and Diet Quality Among Ecuadorian Adults During the COVID-19 Pandemic. A Cross-Sectional Study. Front Nutr. 2022;9(July):1–7. 

5. Rodas JA, Jara-Rizzo MF, Greene CM, Moreta-Herrera R, Oleas D. Cognitive emotion regulation strategies and psychological distress during lockdown due to COVID-19. Int J Psychol. 2022;57(3):315–24. 

6. Bates BR, Moncayo AL, Costales JA, Herrera-Cespedes CA, Grijalva MJ. Knowledge, Attitudes, and Practices Towards COVID-19 Among Ecuadorians During the Outbreak: An Online Cross-Sectional Survey. J Community Health [Internet]. 2020;45(6):1158–67. Available from: https://doi.org/10.1007/s10900-020-00916-7

7. Hidalgo-Andrade P, Hermosa-Bosano C, Paz C. Teachers’ mental health and self-reported coping strategies during the covid-19 pandemic in ecuador: A mixed-methods study. Psychol Res Behav Manag. 2021;14(July):933–44.

 

ANSWERS TO REVIEWER #2:

Reviewer #2: The study titled "Determinants of self-reported health status during COVID-19 lockdown among surveyed Ecuadorian population: a cross sectional study" is written very well. I just have two major questions, that needs clarification to enhance the manuscript.

First, the study conducted backed in July 2020 to October 2020, mean two years back, they why the findings are reported so late, there should an explanation for this in the manuscript.

Thank you very much for your positive feedback. Regarding your comment, it is true that the manuscript is arriving to a scientific journal two years after conducting the survey. The scientific process, including conceptualization, conceptual framework construction, database management, analyses and reflexive review of results can sometimes take long times. Our international team has taken time to review data and the manuscript thoroughly. However, it is important to mention that preliminary results were presented to local authorities in order to better respond to this, and other, health emergencies in the future. We have also used our research results to advocate for improvements in guidelines, regulations, and policies around mental and self-reported health in our country. 

Finally, we believe that even though the COVID-19 pandemic has eased, the impact from the lockdown should be a permanent concern at a global level because of the continuous waves of COVID-19 facing diverse territories around the world, and because of the possibility of new pandemics emerging in a near future. In that regard, our results have value not only to improve current responses to COVID-19, but also to inform future interventions during health emergencies. Our study shows that health authorities and policy makers should consider the potential harms of restrictive measures such as lockdowns to contain disease dissemination. Authorities should also consider applying interventions to reduce risk factors of bad mental and self-reported health. Our manuscript also presents important findings regarding the relevance of including social, environmental and gender considerations in the design of health interventions. 

Second, the authors stated that "Their median (IQR) age was 34 (27–44) years, most 33 participants had a university education (84%) and a full-time public or private job (63%); 34 16% of participants had poor health self-perception". This clearly indicated that there is selection bias in the study participants. For example, in a country as per the manuscript " In the same year, 24.2% of the urban population and 49.2% of the rural population lived under the poverty line [16]. Moreover, only approximately 30% of the economically active population had an adequate job...", where the genral population is 30% economically active, and the study sample size is 63% full-time job, this is clearly selction-bias. So the authors should take into consideration this poin and revised the mansucripts mention this part in limittaions or methods. As per this comment the whole analysis and discussion part needs to be revised.

Thank you for your accurate comment. As it happened with many studies during lockdown, an electronic survey had to be applied to obtain information in that setting. The advantage of this approach was that relevant information could be obtained in a time were performing research had many challenges. The disadvantage was that access to the survey could be limited to some populations (higher socioeconomic status, younger age, etc.). The manuscript is careful stating this composition of the sample and never tries to generalize conclusions. Nevertheless, it could be assumed and discussed (using all the formed body of evidence on health inequalities and social determinants of health) that associations and inequalities found in this study could be even greater if more vulnerable populations would have participated more in the survey.

Also, as we stated in the methods section (lines 153-254), participants were recruited through online platforms and social media using convenience and snowball sampling. This recruitment technique explains the high economic and educative level of the population reached. We also explain how we approached this issue in the limitations section of the manuscript (lines 498-514). 

Also, we have added next text in the discussion section (subsection of limitations), lines 462 to 466:

“As it happened with many studies during lockdown, an electronic survey had to be applied to obtain information in that setting. The advantage of this approach was that relevant information could be obtained in a time were performing research had many challenges. The disadvantage was that access to the survey could be limited to some populations (higher socioeconomic status, younger age, etc.).”

 

ANSWERS TO REVIEWER #3:

Reviewer #3: The study duration in the abstract mentions July to October 2020, while the introduction states that it was conducted between March to October 2020.

The article states “We conducted a cross-sectional study of adults living in Ecuador between July and October 2020”. Does this imply the duration of the study or that the adults were in Ecuador during this time period? This point needs to be made clear in the abstract and introduction.

Thank you very much for noticing this discrepancy. We have clarified that the survey was conducted between July to October 2020, nevertheless, we included in the study participants who lived in Ecuador during the period between March to October 2020. These facts have been clarified in the Abstract (lines 28 to 29, page 2), as follows:

“We conducted a cross-sectional survey of between July and to October 2020 to adults who were living in Ecuador between March to October 2020.”

Furthermore, we added next text in the last part of the Introduction section, lines 144 to 146, page 6:

“Therefore, and in order to examine the association between self-reported health status and its associated factors during Ecuador’s COVID-19 lockdown, we conducted a cross-sectional survey of between July and to October 2020 to adults who were living in Ecuador between March to October 2020.”

Table 1 and Table 2 have missing data. For example, Table 1 mentions there are 1801 females but does not mention males. In addition, the Education level section mentions that 408 individuals do not have university level education while 2191 individuals do. The total number (2191+408) does not equal to 2924 which is the total sample size. This mistake is evident throughout Table 1 and Table 2.

Thanks a lot for your accurate revision. We agree that is important to clarify the gaps you mentioned. In order to clarify the difference between the numbers shown in the tables and the number of the total sample size we have added a new S1 Table called “S1Table, and we have reordered the rest of the supplementary tables. Description of the sample and missing values per each variable”. In agreement with that, we have added next text in the foot of the Table 1 and Table 2: 

“a = There were missing data (<23%) in some variables. For further details, see S1 Table.”

You can now corroborate that the number of the total sample is the resulting from 2191 (university level education) + 408 (lower than university education) +325 (missing values) = 2494. 

Regarding missing values, we have modified a little that missing data was managed by a “complete case analyses”, as follows (lines 212 to 213 of the 9th page, “Statistical analyses and sample considerations” section):

“Considering that the percentage of missing data was <23%, we employed a complete case analysis to estimate statistical associations (for further details see S1 Table).”

Importantly, other authors consider a percentage of missing values per variable of no more than 30% to be acceptable (8).

Reference: 

8. Rubin DB. Multiple Imputation after 18 + Years Multiple Imputation After 18 + Years. J Am Stat Assoc. 2012;91(434):473–89.

The paragraph below Table 1 mentions moderate anxiety levels to be 76% while the Table 1 states that it is 25%. The paragraph below Table states “Severe anxiety and depression were present in 17% of women and 12% of men.” Where are these percentages taken from as Table 2 includes anxiety and depression as two different variables with different percentages between males and females. How were the percentages of 17 and 12 calculated? Table 1 and Table 2 have rounded percentages. However, the paragraph below Table 1 includes percentages such as 38.1%, 29.3%, 35.4%,26.5%. These percentages are not mentioned in Table 1 or Table 2.

Thank you very much for your accurate observations regarding the numbers we showed in the tables; we apologize for the lack of clarity showing those results. The percentage of 76% makes reference to the percentage of “any anxiety level”, and the 69% makes reference to the percentage of “any depression level”. We have corrected both, Table 2 and the paragraph as follows.

In lines 238 to 241 of the 12th page, we have changed the text in this way: 

“Sixteen percent of the participants had regular or poor health self-perception status. In the whole sample, the prevalence of severe anxiety was 17% and severe depression was 7%; nevertheless, 76% had any anxiety level and 69% had any depression level (Table 1).”

In table 2, for clarity, and in order to make it easier to compare anxiety and depression with results obtained from other studies, we only included the category of “moderate to severe anxiety (10 to ≥15 points in the GAD-7 questionnaire)” rather the two categories “moderate anxiety (10 to <15 points)” and “severe anxiety (≥15 points)”. Also, we only included the category of “moderate to severe depression (10 to ≥15 points in the PHQ-9 questionnaire)” rather the three categories “moderate depression (10 to <15 points)”, “moderately severe depression (15 to <20 points), and “severe depression (≥15 points)”.

Furthermore, in lines 241 to 244 of the 12th page, we have changed the text in this way:

“When comparing between both sex categories (Table 2), moderate to severe anxiety levels (10 to ≥15 points of the GAD-7 questionnaire) were reported in 38% of the women and 29% of the men; and, moderate to severe depression levels (10 to ≥20 points of the PHQ9 questionnaire) were reported in 35% of the women and 26% of the men.”

In the Discussion section, lines 389 to 399 of the 24th page, we have corrected the mistakes in the percentages of moderate to severe anxiety and depression as follows:

“In addition, we found that 38% of women and 29% of men reported moderate to severe anxiety, and moderate to severe depression levels were reported in 35% of women and 26% of men. We corroborated that there were gender differences in depression and anxiety, as well as differences in the quantity of (subclinical) depressive symptoms [19]. Specifically, the frequency of anxiety and depression levels was greater in females than in males in the study population. Prior studies found that the levels of such mental health problems were much lower than our findings. Specifically, in one study in Spain [27], 31% of women and 18% of men reported moderate to severe anxiety, and moderate to severe depression levels were reported in 29% of women and 17% of men. It is possible that specific contexts determine reactions to the lockdown. We believe that the Ecuadorian population felt a profound health impact owing to the poor and uncoordinated pandemic response from national and local authorities.”

Please note that we have rounded the percentages as it has been done in the rest of the text and tables.

The article seeks to demonstrate that males and females are both affected with females being more affected. The authors then discuss the different determinants affecting both males and females. However, the discussion section only mentions “being female” as a determinant of “poor self-reported health status”. The conclusion section also makes specific mention of “being female” as a determinant and goes on to ignore males who were discussed in depth in the preceding section.

Thank you very much for your insightful comments. First, thanks to your observation, we realized that table 4 of the manuscript actually corresponded to S1 table (results of sensitivity analysis). We apologize for that. For this reason, we have proceeded to correct the manuscript, placing the correct table 1, which contains the multivariate analyzes that support the results that appear in the main text of the results section. We also changed the order of writing the determinants of poor to self-reported health so that it is consistent with the order in which these results appear in Table 4. The new text, in lines 258-271 reads as follows:

“The multivariate analyses (Table 4) showed that being female (aOR=1.5, 95% CI:1.1 to 2.2), having solely public healthcare system access (aOR=1.9, 95% CI: 1.2 to 2.9), perceiving housing as inadequate to cope with lockdown (aOR=2.2, 95% CI:1.4 to 3.4). Furthermore, perceiving very high difficulties in coping with work or managing household chores was associated with poor health self-perception (aOR=2.7, 95% CI:1.5 to 5.0). The odds of poor self-reported health status were as high as the increase in the number of cohabitants who required care (aOR=1.2, 95% CI:1.1 1.3). Furthermore, having had a diagnosis of COVID-19 or having had COVID-19 symptoms (aOR=3.1, 95%CI:2.0 - 4.7), and suffering from chronic diseases (aOR=6.9, 95% CI:4.9 to 9.7), having severe depression (aOR 5.9, 95%CI:3.1-11.2), were independently associated with poor health self-perception; specifically, there was a “dose-response” association between increasing depression severity and regular or poor self-perception of health; specifically, there was a 60% (65% CI:59%–83%, p-for-trend <0.001) increase in the odds of poor self-reported health status for each change to a higher depressive category.”

Second, certainly, the discussion section lacks clarity regarding a proper analysis in terms of being male and the effects from lockdown. Males and females were affected, but females had a higher proportion of poor to regular self-reported health. Despite males and females were affected in terms of a higher proportion of poor self-reported health status and of severe to moderate anxiety and depression, it seems that females were more affected. 

In order to improve the Discussion section, by a deeper analysis about the impact of lockdown on men, we have added next paragraphs:

Lines 343 to 346, page 22nd:

“Although it is true female had a poorer perception of self-reported health; both, men and women suffered a significant impact on their health, especially in the area of mental health. The fact that we found a worse self-perceived health in women compared to men is coincident with the results obtained from previous studies (9,10).” 

Lines 376 to 381, page 24th:

“Second, according to previous studies (11), potential explanations of the accentuation of poor self-reported health by depression are related with the fact that the presence of symptoms of depression have been associated with the occurrence of severe sexual functioning disorders; and third, the intensity of other fears, not measured in this survey, could be correlated with depressive and sexual disorders.”

Finally, we have changed the order of several paragraphs as it could be seen in the new version of the manuscript (tracked changes version).

References:

9. Czymara, C. S., Langenkamp, A., & Cano, T. (2021). Cause for concerns: gender inequality in experiencing the COVID-19 lockdown in Germany. European Societies, 23(S1), S68–S81. https://doi.org/10.1080/14616696.2020.1808692.

10. Dang, H. A. H., & Viet Nguyen, C. (2021). Gender inequality during the COVID-19 pandemic: Income, expenditure, savings, and job loss. World Development, 140, 105296. https://doi.org/10.1016/j.worlddev.2020.105296

11. Szuster, E., Pawlikowska-Gorzelańczyk, A., Kostrzewska, P., Mandera-Grygierzec, A., Rusiecka, A., Biernikiewicz, M., … Kałka, D. (2022). Mental and Sexual Health of Men in Times of COVID-19 Lockdown. International Journal of Environmental Research and Public Health, 19(22). https://doi.org/10.3390/ijerph192215327

6. PLOS authors have the option to publish the peer review history of their article (what does this mean?). If published, this will include your full peer review and any attached files.

Do you want your identity to be public for this peer review? For information about this choice, including consent withdrawal, please see our Privacy Policy.

Reviewer #1: No

Reviewer #2: Yes: Junaid Ahmad

Reviewer #3: No

Thank you, we did not find any attachment file.

Thank you, we did not build any figure.

---

## [Decision Letter · Decision Letter 1]

21 Feb 2023

Determinants of self-reported health status during COVID-19 lockdown among surveyed Ecuadorian population: a cross sectional study

PONE-D-22-26130R1

Dear Dr. Iván Dueñas-Espín,

We’re pleased to inform you that your manuscript has been judged scientifically suitable for publication and will be formally accepted for publication once it meets all outstanding technical requirements.

Kind regards,

Hadi Ghasemi

Academic Editor

PLOS ONE

Additional Editor Comments (optional):

Reviewers' comments:

Reviewer's Responses to Questions

**Comments to the Author**

1. If the authors have adequately addressed your comments raised in a previous round of review and you feel that this manuscript is now acceptable for publication, you may indicate that here to bypass the “Comments to the Author” section, enter your conflict of interest statement in the “Confidential to Editor” section, and submit your "Accept" recommendation.

Reviewer #2: All comments have been addressed

Reviewer #3: All comments have been addressed

2. Is the manuscript technically sound, and do the data support the conclusions?

Reviewer #2: Partly

Reviewer #3: Yes

3. Has the statistical analysis been performed appropriately and rigorously? 

Reviewer #2: Yes

Reviewer #3: Yes

4. Have the authors made all data underlying the findings in their manuscript fully available?

Reviewer #2: Yes

Reviewer #3: Yes

5. Is the manuscript presented in an intelligible fashion and written in standard English?

Reviewer #2: Yes

Reviewer #3: Yes

6. Review Comments to the Author

Reviewer #2: Thank you for revising the manuscript titled "Determinants of self-reported health status during COVID-19 lockdown among surveyed Ecuadorian population: a cross sectional study". I think the you have addressed all the comments very well. I don't have any further comments or suggestions..

Reviewer #3: (No Response)

7. PLOS authors have the option to publish the peer review history of their article (what does this mean?). If published, this will include your full peer review and any attached files.

Reviewer #2: **Yes: **Junaid Ahmad

Reviewer #3: No

---

## [Editor Report · Acceptance letter]

27 Feb 2023

PONE-D-22-26130R1 

Determinants of self-reported health status during COVID-19 lockdown among surveyed Ecuadorian population: a cross sectional study 

Dear Dr. Dueñas-Espín:

I'm pleased to inform you that your manuscript has been deemed suitable for publication in PLOS ONE. Congratulations! Your manuscript is now with our production department. 

Kind regards, 

on behalf of

Dr. Hadi Ghasemi 

Academic Editor

PLOS ONE